# Associations between police harassment and distrust in and reduced access to healthcare among Black sexual minority men: A longitudinal analysis of HPTN 061

Jonathan P. Feelemyer[1]*, Dustin T. Duncan[2], Molly Remch[3], Jay S. Kaufman[4], Charles M. Cleland[1], Amanda B. Geller[5], Typhanye V. Dyer[6], Joy D. Scheidell[1], Rodman E. Turpin[6], Russell A. Brewer[7], Christopher Hucks-Ortiz[8], Medha Mazumdar[1], Kenneth H. Mayer[9], Maria R. Khan[1]

1 Department of Population Health, New York University School of Medicine, New York, NY, United States of America, 2 Department of Epidemiology, Mailman School of Public Health, Columbia University, New York, NY, United States of America, 3 UNC Gillings School of Global Public Health, Chapel Hill, NC, United States of America, 4 Department of Epidemiology, Biostatistics & Occupational Health, McGill University, Montreal, QC, United States of America, 5 Department of Criminology, Law and Society at the University of California, Irvine, CA, United States of America, 6 Department of Epidemiology and Biostatistics, University of Maryland School of Public Health, College Park, MD, United States of America, 7 Department of Medicine, University of Chicago, Chicago, IL, United States of America, 8 Black AIDS Institute (BAI), Los Angeles, CA, United States of America, 9 The Fenway Institute, Fenway Health and Department of Medicine, Beth Israel Deaconess Medical Center/Harvard Medical School, Boston, MA, United States of America

* jf3880@nyu.edu

## Abstract

### Objective

Evaluate associations between racialized and homophobia-based police harassment (RHBPH) and healthcare distrust and utilization among Black Sexual Minority Men (BSMM).

### Methods

We utilized data from a longitudinal cohort study from HIV Prevention Trials Network (HPTN) 061 with baseline, six and 12 month follow-up assessments. Using multivariable analysis, we evaluated associations between RHBPH and healthcare distrust and utilization reported at the 6 and 12 month visits.

### Results

Of 1553 BSMM present at baseline, 1160 were available at six-month follow-up. In multivariable analysis, increasing frequency of RHBPH was associated with increasing levels of distrust in healthcare providers (aOR 1.31, 95% CI: 1.00, 1.74) and missing 50% or more of healthcare visits at six-month follow-up (aOR 1.93, 95% CI: 1.09, 3.43).

### Conclusions

Recent experiences of RHBPH are associated with reduced trust in and access to healthcare among BSMM, with more frequent RHBPH associated with greater vulnerability.

**Data Availability Statement:** Access to the data can be provided through an approved Data Use Agreement between our institution (New York University School of Medicine) and the institution with which the user is affiliated. Persons wanting to access the data should communicate with the NYU IRB to initiate a Data Use Agreement (irb-info@nyulangone.org).

**Funding:** • Cooperative Agreements UM1 AI068619, UM1 AI068617, and UM1 AI068613, National Institute of Allergy and Infectious Disease (NIAID), National Institute on Drug Abuse (NIDA) and National Institute of Mental Health (NIMH) • CFAR (P30 AI060354) and CTU for HIV Prevention and Microbicide Research (UM1 AI069480), Fenway Institute Clinical Research Site (CRS): Harvard University • CFAR (P30 AI087714, George Washington University CRS: District of Columbia Developmental • CTU (5U01 AI069466) and ARRA funding (3U01 AI069466-03S1), Harlem Prevention Center CRS and NY Blood Center/Union Square CRS: Columbia University • CTU (5U01 AI069418), CFAR (P30 AI050409) and CTSA (UL1 RR025008), Hope Clinic of the Emory Vaccine Center CRS and The Ponce de Leon Center CRS: Emory University HIV/AIDS • CRS: ARRA funding (3U01 AI069496-03S1, 3U01 AI069496-03S2), San Francisco Vaccine and Prevention • CTU (U01 AI069424), UCLA Vine Street CRS: UCLA Department of Medicine, Division of Infectious Diseases • R01DA044037, National Institute on Drug Abuse grant 'Stop-and-Frisk, Arrest, and Incarceration and STI/HIV Risk in Minority MSM (MK) • P30 DA011041, New York University Center for Drug Use and HIV Research • U48 DP005008, New York University-City University of New York (NYU-CUNY) Prevention Research Center • U48 DP006382, University of Maryland Prevention Research Center • P30DA027828-08S1, National Institute on Drug Abuse (NIDA) The funders had no role in study design, data collection and analysis, decision to publish, or preparation of the manuscript.

**Competing interests:** The authors have declared that no competing interests exist.

## Introduction

Sexual minorities face elevated risk of numerous negative health outcomes: mental illness including depression and anxiety [1,2], substance use risk including overdose [3], and blood-borne infection including HIV [4]. Black sexual minority groups face disproportionate risk of many health outcomes compared with white sexual minority counterparts. For example, an estimated 50% of Black sexual minority men versus one in 11 white sexual minority men will be diagnosed with HIV in their lifetime, while rates of mental disorders are also disproportionately high in Black sexual minority groups [5]. Being a racial/ethnic minority as well as being sexual minority have been shown to be associated with psychosocial vulnerability and adverse mental health outcomes, in which network and structural factors are implicated [6]. Given the substantial health challenges experienced by Black sexual minority groups, there is a critical need to ensure adequate access to healthcare to ensure screening, and treatment and prevention to address mental health, substance use and related infectious disease risk.

Black people, and Black sexual minorities in particular, do not have the same financial access to healthcare care as their White counterparts; over 50% of sexual minorities report employment discrimination [7], which can lead to loss of jobs and income, a loss of health insurance, and the inability to access quality healthcare [8]. This financial instability results in a lack of primary care utilization and reliance on emergency department utilization [9]. However, access to quality care refers not only to the ability to afford and logistically access quality healthcare, but to a perception that care is trustworthy, socially and culturally appropriate, and free from stigma and discrimination [10]. There is a longstanding history of racism and discrimination faced by racial/ethnic minorities [11] and sexual minorities in healthcare settings, with many early studies documenting the negative attitudes doctors had with respect to treating sexual minority patients [12,13]. Those at the intersection of racial/ethnic and sexual minority status appear to experience disproportionate history of racial and sexual identity discrimination that results in healthcare distrust; a study conducted in a sample of 500 Black sexual minority men noted that approximately 29% reported racial and sexual identity stigma from their healthcare providers, and 48% reported mistrust in medical establishments including healthcare settings [14]. Among Black sexual minority men, stigma and mistrust both prevent utilization of care and, if care is sought, reduce disclosure of sexual minority status, which has critically important deleterious effects on the quality of care that can be provided [15]. Despite the disproportionate health burden faced by Black sexual minority populations, inadequate access to quality care, and care-related racism and sexual identity discrimination that hinders care in this group, research on social and structural factors that may contribute to these care outcomes is limited.

Police harassment, specifically racialized and homophobia-based police harassment (RHBPH) is a highly prevalent structural determinant of health among Black sexual minorities, and in particular Black sexual minority men (BSMM), that has remained largely unexamined as a potential driver of distrust in healthcare and hindered access to primary care. Racial minority populations are at an elevated risk of RHBPH, with reports highlighting racial disparities in both police stops and arrests [16–20]. Although less studied, individuals with both racial and sexual minority status may be at particularly high risk of aggressive police contact [21].

BSMM experience elevated rates of RHBPH for a number of reasons. First, racial minority populations face disproportionate levels of discrimination in the criminal justice system, leading to higher rates of police encounters [22], Second, compared to their white counterparts, minority populations who face police encounters are more likely to experience aggressive physical contact and harsh language from law enforcement [23,24], and there is evidence that sexual minorities are treated more disrespectfully when stopped by police, possibly due to

transgressions in gender norms [25]. Among sexual minority populations, nearly half have had encounters with police have reported police misconduct, which can lead to stress and mental health problems among these populations [26].

Aggressive policing places subjects at risk of violent outcomes; while Black men make up only 6% of the US population, strikingly, they accounted for 33% of the unarmed individuals killed by police in 2016 [27]. Highly visible examples of police violence in the US among Black individuals have heightened awareness of policing as a threat to both health and driver of racial disparities in health. Police encounters can be a traumatic experience, and often result in negative health outcomes including physical injury, mental trauma, and death [28]; several recent events, including the 2020 killing of George Floyd, have highlighted these negative effects [29–32]. Additionally, it has been shown that for those that experience broader experiences of racism and sexual identity discrimination outside of the healthcare setting, these experiences can directly lead to mistrust in the healthcare settings [33]. These negative social experiences which include RHBPH can not only shape health status, but also may influence healthcare encounters, including mistrust among doctors and clinicians in healthcare settings [34].

Violent police encounters may necessitate emergency department care [20,35], and individuals may choose to visit the ED rather than their primary doctor given that formal records of patients history are not likely to be as prevalent in ED settings as compared to primary care doctor offices. They may also visit the ED because they may not have a primary doctor to rely on for care or due to lack of health insurance. However, even when emergency care is not immediately required, the specter of RHBPH in communities may exert strain and stress, resulting in distrust in care systems. Sociological theory has suggested police contact results in "systems avoidance" and withdrawal from institutions [36], which may include avoiding medical doctors, clinics, or other outpatient settings where formal chart records are kept. Related work examining immigration enforcement suggests that individuals at high risk of law enforcement contact may withdraw from their social networks [37]. To the extent that system avoidance keeps justice-involved individuals away from healthcare institutions or more informal support, they may be forced to rely on emergency departments for health needs that could otherwise have been treated in primary care settings. It should be noted, for instance, that excessive police violence may lead to individuals visiting the emergency department, where information would be collected on the circumstances of the injuries and may lead to further stigmatization of the individuals who experience these police related injuries [38,39].

The objective of this study is to assess associations between RHBPH and subsequent distrust in care, interruptions in primary care as indicated by missed visits and reliance on the emergency room for care among BSMM, with specific focus on RHBPH perceived to be motivated by their race and/or sexual identity. Understanding how this structurally enforced police harassment may influence trust in healthcare systems, access to primary care healthcare use, and use of emergency department care is important to assessing how to reduce these challenges and the subsequent disparate negative health outcomes.

## Materials and methods

### Study design and participants

Data from the HIV Prevention Trials Network 061 (HPTN 061) study were used for the analysis. The study's enrollment and recruitment methods have been described comprehensively elsewhere [40]. HPTN 061 sought to examine the feasibility and efficacy of interventions to prevent the acquisition and transmission of HIV among BSMM. Enrollment took place from 2009–2010 in six US metropolitan cities: Atlanta, Boston, Los Angeles, New York City, San Francisco, and Washington DC.

## Measures and eligibility

Participants were recruited directly from the community or as sexual network partners referred by index participants. Individuals were eligible to participate in the study if they were Black, African American, Caribbean Black, or multiethnic Black; identified as a man or were assigned male at birth; were at least 18 years old; reported any condomless anal intercourse with a man in the prior six months; resided in the metropolitan area of the research clinic; did not plan to move away during the study period; and provided informed consent for the study. Individuals were ineligible if they were enrolled in any other HIV interventional research study, had been a participant in an HIV vaccine trial, or were a community-recruited participant in a category that had already reached its enrollment cap. Pre-screening to determine eligibility was performed either in person or over the telephone.

Institutional review boards at all participating institutions approved the original study. Participants completed an audio computer-assisted self-interview (ACASI) at baseline, six-month, and twelve-month follow-up that assessed demographic information, HIV risk behaviors, and other important psychosocial and health-related characteristics.

## Racialized and homophobia-based police harassment exposure

At the baseline and six-month visits, participants were asked about past six-month experience with police harassment motivated by racism or homophobia. In the survey, participants were asked if they were harassed by police; if they responded yes to this question, they were asked two follow up questions including if the harassment was perceived to be due to their race, or due to their sexuality. The participants were allowed to answer yes to one or both of these questions. We coded their responses into a three level ordinal exposure indicating police harassment due to either racism or homophobia at different time points: level 0 indicates no such harassment at either time period (i.e., baseline or the six month follow-up visit), level 1 indicates that the participant reported RHBPH at the baseline visit or at the six month follow-up visit, and level 2 indicates that the participant reported RHBPH at both the baseline and six month follow-up visit.

## Healthcare outcomes

At the six and 12 month visits, participants were asked several questions about their access to healthcare and perceived stigma among providers in the six months prior to the follow-up visit, including whether the participant had seen their healthcare provider and if so, if they had missed 50% or more of their scheduled appointments (information on missed visits was only collected for those that had reported seeing a healthcare provider; for those that did not report seeing their healthcare provider, this question was skipped in the survey for those individuals). Participants were also asked if they had seen their healthcare provider at any point in the last six months, and if so, they were asked if they reported any mistrust in their doctor/healthcare provider (similar to the missed visits variable, this question was only asked for those that had reported seeing their healthcare provider and was skipped for those who had not reported a recent healthcare provider visit). Finally, participants were asked if they had accessed the emergency department for any treatment (given that those without a reliable provider or not accessing their provider/primary care doctor could be accessing emergency room care instead, particularly for those requiring immediate care that cannot be provided by a primary care doctor; they may also choose to visit the ED given their mistrust with their primary care doctors). Participants responded to each of these questions with a yes or no response.

## Covariates

Baseline covariates selected a priori based on prior research [40–42] included: age, transgender status (measured as yes/no), insufficient income (reported by participant as not having sufficient income in the previous six months) (measured as yes/no), BSMM who also have sex with women (measured as yes/no), unstable housing (did not have a stable place of housing based on self-report) (measured as yes/no), education (measured as greater than higher school vs less than high school), city of residence, HIV serostatus, history of any STI infection (including Syphilis, Gonorrhea, or Chlamydia infection based on laboratory testing) (measured as yes/no), depression (measured by the CES-D questionnaire [43]) and baseline reporting of each of the outcome healthcare access variables (measured as yes/no) and having health coverage (including any type of health insurance coverage) (yes/no).

## Statistical analysis

The prevalence of baseline demographic information, RHBPH frequency, and healthcare distrust and utilization indicators were calculated for the analytic sample. We used logistic regression to estimate unadjusted and adjusted odds ratios (aORs) and 95% confidence intervals (CIs) for associations between frequency of RHBPH over follow-up (reported at the baseline to 6-month visits) and healthcare distrust and utilization outcomes measured at the 6- and 12-month visits. Multivariable models controlled for baseline experience of healthcare access, multiple demographic, and self-report behavioral characteristics (a priori variables, listed in Table 1 and described above). Stata 16 [44] was used for statistical analysis.

# Results

## Demographic characteristics

Of 1,553 participants in the baseline sample, 1,160 who responded to the questions on RHBPH at the baseline and six-month visit were included in the twelve-month follow-up analysis. Participant demographics are described in Table 1. At baseline, the majority of participants were non-Hispanic Black and unemployed; approximately 47.3% had at least a high school education, over 55.4% made less than $20,000 per year, 9.5% reported unstable housing, and 60.1% reported a lifetime history of incarceration. A total of 43.2% reported a history of depression. We did not identify significant differences between those included and those not included in the follow-up sample.

Table 2 provides prevalence of each of the healthcare outcomes at six and twelve months stratified by different levels of RHBPH (i.e. none, RHBPH at baseline or six months, RHBPH at baseline and six months). At the six-month follow-up visit, 25.9% to 33.3% of those reporting select healthcare outcomes reported RHBPH at baseline or six months and 57.7% to 67.5% experienced RHBPH at both baseline and six months. At the 12-month follow-up visit, those reporting select healthcare outcomes reported RHBPH ranging from 27.7% to 36.0% for those who experienced RHBPH at baseline or six months and 54.4% to 66.5% among those who experienced RHBPH at both baseline and six months.

## Association between police harassment with healthcare outcomes

Table 3A presents the bivariate and multivariate models associations between frequency of RHBPH over cohort follow-up and healthcare related outcomes reported at the 6-month follow-up visit. Approximately 89.8% reported some form of RHBPH at the baseline or six-month visit, and approximately 57% reported RHBPH at both visits. Increasing frequency of RHBPH over follow-up was associated with distrust in healthcare providers (OR: 1.46, 95% CI:

**Table 1. Sample demographic, socioeconomic and health background among baseline and follow-up, HPTN 061.**

| | | Total Sample N | Total Sample % | 12-month Follow-up N | Follow-up % |
|---|---|---|---|---|---|
| | | 1553 | 100 | 1160 | 74.7 |
| **Age** | | | | | |
| | 18–30 | 517 | 33.4 | 396 | 34.2 |
| | 31–50 | 812 | 52.4 | 604 | 52.1 |
| | 50 and over | 220 | 14.2 | 159 | 13.7 |
| **Ethnicity** | | | | | |
| | Non-Hispanic | 1430 | 92.3 | 1069 | 92.2 |
| | Hispanic | 119 | 7.7 | 90 | 7.8 |
| **Education** | | | | | |
| | Greater than High School | 732 | 47.3 | 596 | 51.5 |
| | High School | 816 | 52.7 | 562 | 48.5 |
| **Insufficient Income** | | | | | |
| | No | 690 | 44.6 | 512 | 44.2 |
| | Yes | 858 | 55.4 | 647 | 55.8 |
| **Transgender** | | | | | |
| | No | 1485 | 95.9 | 1106 | 95.9 |
| | Yes | 63 | 4.1 | 47 | 4.1 |
| **Men who have sex with men and women** | | | | | |
| | No | 872 | 56.3 | 782 | 67.7 |
| | Yes | 676 | 43.7 | 373 | 32.3 |
| **Unstable Housing** | | | | | |
| | No | 1401 | 90.5 | 1046 | 90.3 |
| | Yes | 148 | 9.6 | 113 | 9.7 |
| **City of Residence** | | | | | |
| | Washington DC | 227 | 14.6 | 177 | 15.3 |
| | Atlanta | 292 | 18.8 | 206 | 17.8 |
| | Boston | 237 | 15.3 | 171 | 14.8 |
| | Los Angeles | 283 | 18.2 | 203 | 17.5 |
| | New York City | 310 | 19.9 | 253 | 21.8 |
| | San Francisco | 204 | 13.1 | 150 | 12.9 |
| **Health Coverage** | | | | | |
| | No | 613 | 39.6 | 451 | 38.9 |
| | Yes | 936 | 60.4 | 708 | 61.1 |
| **History of Incarceration** | | | | | |
| | No | 607 | 39.9 | 464 | 40.7 |
| | Yes | 914 | 60.1 | 676 | 59.3 |
| **HIV Serostatus** | | | | | |
| | Negative | 1167 | 77.1 | 927 | 81.3 |
| | Positive | 348 | 22.9 | 213 | 18.7 |
| **STI (Includes a diagnosis of Chlamydia, Syphillis or Gonorrhea)** | | | | | |
| | Negative | 1245 | 86.3 | 943 | 86.9 |
| | Positive | 198 | 13.7 | 142 | 13.1 |
| **Depression** | | | | | |
| | No | 759 | 56.8 | 588 | 58.1 |
| | Yes | 578 | 43.2 | 424 | 41.9 |

**Table 2. Levels of racialized and homophobia-based police harassment and distribution of healthcare related outcomes at six and twelve month visits among HPTN 061 participants.**

| | Has Seen Healthcare Provider | Went to the Emergency Room | Missed 50% or more of Healthcare Visits | Trust the health care providers I see in my health care setting |
|---|---|---|---|---|
| **Six Month Follow-up Visit** | | | | |
| | N (%) | N (%) | N (%) | N (%) |
| No racialized and homophobia-based police harassment | 67 (8.9) | 17 (7.9) | 5 (6.49) | 16 (7.3) |
| Any racialized and homophobia-based police harassment at baseline or at six months | 250 (33.3) | 59 (27.6) | 20 (26) | 57 (25.9) |
| Any racialized and homophobia-based police harassment at baseline and six months | 433 (57.7) | 138 (64.5) | 52 (67.5) | 147 (66.8) |
| **Twelve Month Follow-up Visit** | | | | |
| | N (%) | N (%) | N (%) | N (%) |
| No racialized and homophobia-based police harassment | 60 (8.6) | 10 (5.1) | 7 (10.8) | 11 (6.5) |
| Any racialized and homophobia-based police harassment at baseline or at six months | 253 (36.0) | 56 (28.4) | 18 (27.7) | 52 (30.6) |
| Any racialized and homophobia-based police harassment at baseline and at six months | 389 (55.4) | 131 (66.5) | 40 (61.5) | 107 (62.9) |

1.15, 1.86), utilizing the emergency room (OR: 1.31, 95% CI: 1.00, 1.74), and missing 50% or more of healthcare visits (OR: 1.50, 95% CI: 1.01, 2.23) at the six-month follow-up visit. In multivariate analysis, associations with distrust in healthcare providers (aOR 1.31, 95% CI: 1.00, 1.74) and missing 50% or more of healthcare visits (aOR 1.93, 95% CI: 1.09, 3.43) weakened yet associations generally remained, and the association with emergency room visitation was further attenuated (aOR: 1.17, 0.82, 1.67) There was minimal evidence of an association between RHBPH and having attended less than 50% of visits to their healthcare provider.

Table 3B presents the bivariate and multivariate models examining frequency of RHBPH over cohort follow-up and healthcare related outcomes reported at the 12-month follow-up visit. Increasing frequency of RHBPH was associated with increasing levels of distrust in healthcare providers (OR 1.36, 95% CI: 1.04, 1.78) and of emergency room utilization, (OR 1.69, 95% CI: 1.28, 2.25) in the unadjusted analysis. In adjusted analysis, these associations were attenuated (having distrust in healthcare providers: aOR 1.19, 95% CI: 0.89, 1.61; utilizing

**Table 3. A. Relationships with racialized and homophobia-based police harassment and 6 month healthcare outcomes in HPTN 061.** B. Relationships with racialized and homophobia-based police harassment and 12 month healthcare outcomes in HPTN 061.

| | N (%) | OR (95% CI) | p | aOR (95% CI) | p |
|---|---|---|---|---|---|
| **Healthcare Utilization** | | | | | |
| Trust the health care providers I see in my health care setting | 229 (19.7) | 1.46 (1.15, 1.86) | 0.002 | 1.31 (1.00, 1.74) | 0.050 |
| Healthcare Utilization: Went to the Emergency Room (any use) | 225 (28.5) | 1.30 (1.01, 1.67) | 0.041 | 1.17 (0.82, 1.67) | 0.393 |
| Healthcare Utilization: Missing more that 50% of healthcare visits | 80 (9.7) | 1.50 (1.01, 2.23) | 0.046 | 1.93 (1.09, 3.43) | 0.025 |
| Healthcare Utilization: Has seen healthcare provider | 790 (66.6) | 1.14 (0.95, 1.37) | 0.156 | 1.09 (0.88, 1.36) | 0.395 |
| | N (%) | OR (95% CI) | p | aOR (95% CI) | p |
| **Healthcare Utilization** | | | | | |
| Trust the health care providers I see in my health care setting | 190 (17.5) | 1.36 (1.04, 1.78) | 0.021 | 1.15 (0.85, 1.54) | 0.362 |
| Healthcare Utilization: Went to the Emergency Room (any use) | 227 (28.6) | 1.69 (1.28, 2.25) | <0.001 | 1.36 (0.94, 1.95) | 0.103 |
| Healthcare Utilization: Missing more that 50% of healthcare visits | 79 (9.5) | 1.14 (0.77, 1.69) | 0.512 | 1.04 (0.61, 1.75) | 0.891 |
| Healthcare Utilization: Has seen healthcare provider | 789 (67.3) | 1.08 (0.89, 1.32) | 0.392 | 1.06 (0.85, 1.33) | 0.629 |

the emergency room: aOR 1.39, 95% CI: 0.97, 2.01). RHBPH was not associated with missed visits or having been seen by a provider.

## Discussion

This study was among the first to measure associations between racialized and homophobia-based police harassment and distrust in healthcare and healthcare utilization in a cohort of BSMM in the United States. While violent encounters with police and other law enforcement are common across numerous US settings [45–48], in this large population-based sample of BSMM recruited from urban areas across the US, RHBPH was a nearly universal exposure: 90% reported RHBPH across cohort follow-up at the baseline or the 6 month visit and nearly 60% reported the experience at both time points. Adjusted analyses suggested increasing exposure to RHBPH was independently associated with elevated prevalence of distrust in healthcare providers and emergency room utilization, an established indicator of reduced access to care.

The current study provides one of the first preliminary analysis to suggest discriminatory policing may contribute to distrust in the healthcare system and reduce engagement with and access in healthcare among BSMM. The study also compliments several studies highlighting that, compared to Whites, racial and ethnic minority populations are more likely to express mistrust in the medical system [49–51]. Future research should attempt to validate the associations seen here and better describe the range of the ways police contact and harassment influence healthcare access of BSMM. If further validated, this body of research would advance policy discussions focused on deleterious consequences of policing on health and would highlight that reducing exposure to aggressive policing among BSMM, a critical action in itself, also may reduce distrust of care systems and therefore improve healthcare access and uptake.

Additional research is needed to better understand pathways linking police harassment and care related factors, particularly healthcare related factors. For example, the potential role of police harassment-related stigma should be further explored. Prior models have highlighted among racial/ethnic and sexual minority groups the significant role that intersectional stigma may play in medical mistrust and reduced access to care [52]. Sexual orientation and/or race based stigma is common; in some samples of sexual minority men, upwards of 40% of those who had recently seen a healthcare provider did not disclose their sexual identity due to stigma, with rates even lower among minority men including Black and Latino MSM [53]. Intersectional stigma is significantly associated with reduced engagement in care such as obtaining recent physical examinations [54].

Our findings suggest that negative encounters with the criminal justice system (including interactions with police) could be a threat to healthcare utilization given the potential for anticipatory stigma: having experienced RHBPH, BSMM may fear that they will be stigmatized or harassed in other settings as well, including, potentially, healthcare settings. They may, accordingly, be less likely to seek care (including preventative care) when needed and appropriate. Negative encounters with the criminal justice system could also threaten healthcare utilization through a socialization mechanism: personal experiences with police have the potential to drive legal cynicism or estrangement [55,56], which may extend to a belief that healthcare providers, like police, are a threat rather than a source of care. Mixed methods research is needed to explore the potential range of ways police harassment could influence access to and uptake of (and adherence to) care.

There are several important limitations of the current study. First, we cannot rule out residual confounding in the association between RHBPH and different healthcare outcomes, including other types of violence. This concern is particularly relevant for our analysis of emergency department use: as hypothesized, RHBPH may lead affected individuals to withdraw from primary care settings, leaving emergency departments as their provider of last resort. However, BSMM with the greatest exposure to aggressive policing may also be more deeply

engaged in activities that place them at risk of adverse health events. We also would note that our exposure captures any police harassment among participants but did not ask about the frequency of harassment; it is likely that some participants experienced more individual instances of harassment compared to others. Finally, we would note that those who did not visit a healthcare provider in follow-up, a number of reasons, including being healthy and not requiring healthcare consultations, effects of police harassment, or factors that were not asked about in our survey could explain these lack of visits. For this reason, future studies should attempt to tease apart reasons for not seeing healthcare providers in greater detail and explore how these different reasons may also play a role in medical mistrust and other healthcare outcomes.

An additional concern was low power in some strata of police harassment to isolate effects of particular harassment forms; most notably, few participants reported they had been harassed by the police due to their sexual orientation alone. This prevented meaningful analysis of associations between homophobic harassment and subsequent healthcare outcomes. Many of the variables collected were based on self-report, and there may have been recall bias or social desirability bias in some of the responses. As this is a sample of predominately BSMM in urban areas, these findings may not be generalizable to non-sexual minority populations and those in non-urban areas.

## Conclusion

Our study found important associations between RHBPH and several healthcare utilization outcomes. These findings have the potential to inform interventions and harm reduction programs that improve access to care in sexual minority populations, which may include cultural competence among doctors and clinical providers serving sexual and racial minority populations [57]. Additionally, interventions to train healthcare providers to ensure awareness of the high prevalence of prior criminal justice involvement among some patient populations, to inform potential screening for criminal justice involvement is suggested, and if appropriate, engagement in discussions about patients' experiences including how prior criminal justice involvement may negatively influence access to and experience in care. Doing so would improve the care environment by acknowledging the prior trauma experienced by a patient. In addition, given shared medical decision making and development of personalized medical plans may increase patients' overall trust in medical care [58], it is critical that healthcare providers in heavily policed communities, serving populations affected by high levels of police exposure, actively engage patients in a non-judgmental and strengths-based manner to increase patient activation, so patients affected by negative experiences of criminal justice involvement take ownership of their healthcare.

Future studies should continue to document the domains in which police harassment may undermine health care use and other health outcomes, to help further inform current policy discussion focused on reforming the police and investing resources into public health rather than criminal justice infrastructures. For example, these findings complement our prior work documenting the association between police harassment and subsequent violent experiences, which may lead to increased healthcare usage for injuries sustained through these violent encounters [59]. By understanding numerous and diverse downstream effects of police harassment we build an empirical base of findings necessary to advocate for changes in criminal justice policies that could reduce health inequities.

## Supporting information

**S1 Appendix. Tables S1-S16: Bivariate and full multivariable models (including all covariates) for Tables 3A and 3B for healthcare outcomes.**
(DOCX)

## Acknowledgments

We are thankful to the following groups who made possible the HPTN 061 study: HPTN 061 study participants; HPTN 061 Protocol co-chairs, Beryl Koblin, PhD, Kenneth Mayer, MD, and Darrell Wheeler, PhD, MPH; HPTN061 Protocol team members; HPTN Black Caucus; HPTN Network Laboratory, Johns Hopkins University School of Medicine; Vaccine and Infectious Disease Division, Fred Hutchinson Cancer Research Center; Statistical and Data Management Center, Statistical Center for HIV/AIDS Research and Prevention; HPTN CORE Operating Center, Family Health International (FHI) 360; Black Gay Research Group; clinical research sites, staff, and Community Advisory Boards at Emory University, Fenway Institute, GWU School of Public Health and Health Services, Harlem Prevention Center, New York Blood Center, San Francisco Department of Public Health, the University of California, Los Angeles, Center for Behavioral and Addiction Medicine, and Cornelius Baker, FHI 360. We are thankful to Sam Griffith, Senior Clinical Research Manager, FHI 360, and Lynda Emel, Associate Director, HPTN Statistical and Data Management Center, Fred Hutchinson Cancer Research Center, for their considerable assistance with HPTN 061 data acquisition and documentation.

## Author Contributions

**Conceptualization:** Jonathan P. Feelemyer, Maria R. Khan.

**Formal analysis:** Jonathan P. Feelemyer, Charles M. Cleland.

**Funding acquisition:** Maria R. Khan.

**Investigation:** Maria R. Khan.

**Methodology:** Jonathan P. Feelemyer, Jay S. Kaufman, Charles M. Cleland, Medha Mazumdar, Maria R. Khan.

**Project administration:** Maria R. Khan.

**Supervision:** Maria R. Khan.

**Writing – original draft:** Jonathan P. Feelemyer, Dustin T. Duncan.

**Writing – review & editing:** Jonathan P. Feelemyer, Dustin T. Duncan, Molly Remch, Jay S. Kaufman, Charles M. Cleland, Amanda B. Geller, Typhanye V. Dyer, Joy D. Scheidell, Rodman E. Turpin, Russell A. Brewer, Christopher Hucks-Ortiz, Kenneth H. Mayer, Maria R. Khan.

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
