## [Decision Letter · Decision Letter 0]

21 Aug 2022

PONE-D-22-12905Associations between Police Harassment and Distrust in and Reduced Access to Healthcare Among Black Sexual Minority Men: A Longitudinal Analysis of HPTN 061PLOS ONE

Dear Dr. Feelemyer,

Thank you for submitting your manuscript to PLOS ONE. After careful consideration, we feel that it has merit but does not fully meet PLOS ONE’s publication criteria as it currently stands. Therefore, we invite you to submit a revised version of the manuscript that addresses the points raised during the review process.

Please note that we have only been able to secure a single reviewer to assess your manuscript. We are issuing a decision on your manuscript at this point to prevent further delays in the evaluation of your manuscript. Please be aware that the editor who handles your revised manuscript might find it necessary to invite additional reviewers to assess this work once the revised manuscript is submitted. However, we will aim to proceed on the basis of this single review if possible. Please address each of the reviewer's comments below, particularly those regarding whether the analysis sufficiently supports the conclusions made.

We look forward to receiving your revised manuscript.

Kind regards,

Hugh Cowley

Staff Editor

PLOS ONE

Journal Requirements:

Reviewers' comments:

Reviewer's Responses to Questions

**Comments to the Author**

1. Is the manuscript technically sound, and do the data support the conclusions?

Reviewer #1: Partly

2. Has the statistical analysis been performed appropriately and rigorously? 

Reviewer #1: Yes

3. Have the authors made all data underlying the findings in their manuscript fully available?

Reviewer #1: No

4. Is the manuscript presented in an intelligible fashion and written in standard English?

Reviewer #1: Yes

5. Review Comments to the Author

Reviewer #1: Thank you for the opportunity to review the manuscript entitled “Associations between Police Harassment and Distrust in and Reduced Access to Healthcare Among Black Sexual Minority Men: A Longitudinal Analysis of HPTN 061.” This manuscript examines identity-based police harassment (IBPH) as a predictor of 1) distrust in, and 2) reduced engagement in healthcare for Black men who have sex with men (BMSM).

This is a very important area of study, and the authors are to be commended for these data.

However, a few concerns lead me not to recommend publication at this time. Most importantly, I am not confident that the results support the general conclusions. Section-specific comments are appended below.

Introduction

The authors do a nice job of contextualizing the study. BMSM face disparities in a variety of health domains. Low engagement with healthcare may be an important explanation for these disparities, with established barriers, such as medical mistrust. These could be addressed if we know more about their sources. As such, the research question: what broader socio-structural factors contributes to those barriers?

However, it would be helpful for the reader to have the relationships between specific variables spelled out and justified.

1) Could the authors elaborate on the reasoning why IBPH is likely highly prevalent for BMSM?

2) Could the authors describe the theoretical link between IBPH and medical distrust? The authors introduce systems avoidance, and that the spectacle of police violence could foster distrust in systems more broadly. At a second-hand level, this explanation makes sense. But, how would this play out at the intra-individual level, such that the actual experience changes intra-individual psychological processes? There is some speculation in the discussion, but it would be helpful to anticipate in the introduction.

Finally, the idea that an emergency department (ED) visit is the less stigmatizing alternative to healthcare provider (HCP) is not spelled out. Why is this engagement with healthcare considered the alternative to PCP visit rather than opting out altogether?

Methods

The authors do a nice job describing the sample, their variables, and their construction. Ordinal predictor variable of IBPH is explained well. However, there is a detail about the dichotomization of the predictor variable for mediation is confusing. Further, mediation analyses are not discussed later in the manuscript.

Relatedly, the conceptual framing of the variables places distrust, HCP non-use, and increased ED use as related outcomes of IBHP. Did the authors examined distrust and non-use as mediators of the association between IBPH and increased emergency room visit?

Most importantly, it is clear from the description in the methods that these outcomes are tiered. Namely, data regarding the outcomes of HCP mistrust and missing visits are only collected from participants who did see a HCP in the past 6 months. This detail should be carried throughout the manuscript, as it does impact the conclusions from the study. What about those who did not see a HCP? Couldn’t this be due to mistrust from IBPH? Could those participants be analyzed as well somehow?

Results

Can the authors provide p-values for all models and regression coefficients?

In unadjusted models, there was evidence that IBPH predicted:

1) at 6M, distrust in HCP, missing more than 50% of appointments, and visiting the ED.

2) at 12M, distrust in HCP, visiting the ED.

However, in adjusted analyses, the confidence intervals for all the predictors in the logistic regressions include 1.0 (except for one predictor), which suggests non-significance of the key relationships of interest. Could the authors report all p-values? These models suggest that the hypothesized relationships are due to various covariates. Can the authors report these full models, as well?

Aside from this point, this sample reported a very high rate of previous incarceration (60.1%). I wonder if this particular factor could underpin the outcomes of interest rather than incidences of IBPH. Did the authors run models with incarceration (60.1%)as a covariate or predictor?

6. PLOS authors have the option to publish the peer review history of their article (what does this mean?). If published, this will include your full peer review and any attached files.

Reviewer #1: No

---

## [Author Response · Author response to Decision Letter 0]

5 Oct 2022

Reviewer #1: Thank you for the opportunity to review the manuscript entitled “Associations between Police Harassment and Distrust in and Reduced Access to Healthcare Among Black Sexual Minority Men: A Longitudinal Analysis of HPTN 061.” This manuscript examines identity-based police harassment (IBPH) as a predictor of 1) distrust in, and 2) reduced engagement in healthcare for Black men who have sex with men (BMSM).

This is a very important area of study, and the authors are to be commended for these data.

However, a few concerns lead me not to recommend publication at this time. Most importantly, I am not confident that the results support the general conclusions. Section-specific comments are appended below.

Introduction

The authors do a nice job of contextualizing the study. BMSM face disparities in a variety of health domains. Low engagement with healthcare may be an important explanation for these disparities, with established barriers, such as medical mistrust. These could be addressed if we know more about their sources. As such, the research question: what broader socio-structural factors contributes to those barriers?

However, it would be helpful for the reader to have the relationships between specific variables spelled out and justified.

Author response: We thank the reviewer for the careful review of our important paper and highlighting key strengths of the analysis. We have addressed the reviewer’s comments below as well as in the revised manuscript and, importantly, feel that the paper has been strengthened as a result. We have detailed the relationships between the different variables further in the introduction and discussions sections of the manuscript, and we have additionally highlighted some of the limitations of some of the outcome variables as well (for instance, the use of ED department outcome variable). We have also drawn on previous literature that has evaluated similar relationships between IBPH and healthcare outcomes to further justify the analysis in the manuscript (see Alang, S., McAlpine, D. D., & Hardeman, R. (2020). Police brutality and mistrust in medical institutions. Journal of racial and ethnic health disparities, 7(4), 760-768)

1) Could the authors elaborate on the reasoning why IBPH is likely highly prevalent for BMSM?

Author response: As suggested by the reviewer, we have added a paragraph in the introduction that has introduced several factors associated with IBPH among racial and sexual minority populations, highlighting the disproportionate rates of police encounters among these populations, and some of the negative outcomes associated with these encounters including increased stress and mental health symptoms. We have included several important references as well that have highlighted these disparities in previous research on these particular populations. 

To summarize we note BSMM experience elevated rates of IBPH for a number of reasons. First, racial minority populations face disproportionate levels of discrimination in the criminal justice system, leading to higher rates of police encounters. Second, compared to their white counterparts, minority populations who face police encounters are more likely to experience aggressive physical contact and harsh language from law enforcement, and there is evidence that sexual minorities are treated more disrespectfully when stopped by police, possibly due to transgressions in gender norms. Among sexual minority populations, nearly half who have had encounters with police have reported police misconduct, which can lead to stress and mental health problems among these populations. 

Here is a list of relevant references that were included in the introduction highlighting this information:

Brunson, R. K., & Weitzer, R. (2009). Police relations with black and white youths in different urban neighborhoods. Urban Affairs Review, 44(6), 858-885. 

Davis, E., Whyde, A., & Langton, L. (2018). Contacts between police and the public, 2015. US Department of Justice Office of Justice Programs Bureau of Justice Statistics Special Report, 1-33. 

Fagan, J., Geller, A., Davies, G., & West, V. (2009). Street Stops and Broken Windows Revisited: The Demography and Logic of Proactive Policing in a Safe and Changing City. Retrieved from New York, NY: 

Geller, A. (2017). Policing America’s children: Police contact and consequences among teens in fragile families. IDEAS Work Pap Ser from RePEc, 1-46. 

Mallory, C., Hasenbush, A., & Sears, B. (2015). Discrimination and harassment by law enforcement officers in the LGBT community. 

2) Could the authors describe the theoretical link between IBPH and medical distrust? The authors introduce systems avoidance, and that the spectacle of police violence could foster distrust in systems more broadly. At a second-hand level, this explanation makes sense. But, how would this play out at the intra-individual level, such that the actual experience changes intra-individual psychological processes? There is some speculation in the discussion, but it would be helpful to anticipate in the introduction.

Finally, the idea that an emergency department (ED) visit is the less stigmatizing alternative to healthcare provider (HCP) is not spelled out. Why is this engagement with healthcare considered the alternative to PCP visit rather than opting out altogether?

Author response: In the revised manuscript, we have provided further information on the theoretical link between IBPH and medical distrust, and highlighted findings from some recent articles that has explored links between police harassment and medical mistrust. For example, in the revised manuscript, we state that “broader experiences of discrimination outside of the healthcare setting (which may include disproportionate police harassment among racial minorities) could be directly associated with mistrust in the healthcare system.” (page 5). They also note that these negative social experiences, which may include IBPH, can shape not only health status, but healthcare encounters including seeing one’s healthcare provider. We have added some context to the introduction that introduces this research and the implication that IBPH has on medical mistrust in the healthcare setting.

With respect to emergency department use, we highlight in the introduction that because primary medical doctors often have charts with extensive patient history, some individuals may opt to utilize ED as a way of “systems avoidance” such that system avoidance keeps justice-involved individuals away from healthcare institutions or more informal support, and may thus cause them to rely on emergency departments for health needs that could otherwise have been treated in primary care settings. As many who are justice involved often have instances of police contact, this would play a role in their overall system avoidance and lead them to utilizing the ED as part of this avoidance of their primary doctors. We have added additional language to note that because ED are not as likely to have full historical medical records for individuals, this may lead persons to choose to visit the ED rather than their regular doctor to avoid the stigma they may have experienced at their primary doctor’s office. However, there still remains some stigma in any type of medical encounter, including ED visits, which is why we chose to explore this outcome in our analysis. Furthermore, there is evidence that if individuals visit the ED as a result of the direct police encounter and violence, may lead to further stigmatization of the individuals who experience these police related injuries, particularly if they visit the ED often due to violence that is a direct result of police encounters.

We drew information from the references included in the manuscript, also noted below:

Alang, S., McAlpine, D. D., & Hardeman, R. (2020). Police brutality and mistrust in medical institutions. Journal of racial and ethnic health disparities, 7(4), 760-768. 

Hammond, W. P. (2010). Psychosocial correlates of medical mistrust among African American men. American Journal of Community Psychology, 45(1), 87-106. 

Methods

The authors do a nice job describing the sample, their variables, and their construction. Ordinal predictor variable of IBPH is explained well. However, there is a detail about the dichotomization of the predictor variable for mediation is confusing. Further, mediation analyses are not discussed later in the manuscript.

Author response: We had previously proposed performing a mediation analysis for this paper but after running several models we did not note strong mediating effects and the explanation of the mediation analysis would have made the paper much longer than the journal would allow, so we therefore decided to remove the mediation analysis and only present the multivariable effects of police harassment and healthcare outcomes. We have removed all mentions of mediation analysis in the updated draft and thank the reviewer for pointing this out in the methods section.

Relatedly, the conceptual framing of the variables places distrust, HCP non-use, and increased ED use as related outcomes of IBHP. Did the authors examined distrust and non-use as mediators of the association between IBPH and increased emergency room visit?

Author response: We performed a separate mediation analysis that examined these factors; the results of this analysis did not show any meaningful indirect effects, with no statistically significant findings in the mediation analysis results. We chose not to report this information in the manuscript and will likely perform a more detailed mediation analysis in a subsequent paper. 

Most importantly, it is clear from the description in the methods that these outcomes are tiered. Namely, data regarding the outcomes of HCP mistrust and missing visits are only collected from participants who did see a HCP in the past 6 months. This detail should be carried throughout the manuscript, as it does impact the conclusions from the study. What about those who did not see a HCP? Couldn’t this be due to mistrust from IBPH? Could those participants be analyzed as well somehow?

Author response: We thank the reviewer for highlighting this aspect of healthcare provider visits. The current methods note that the missed visits variable was only asked of those who had seen a healthcare provider in the last six months. The participants in this study did report several co-morbidities that may lead to needing to see a healthcare provider (such as HIV and STI infection); however, there were also a number of individuals who did not have any of these conditions, and may not have had a need to visit a healthcare provider outside of regular yearly screenings/physicals. In the discussion, we have updated the manuscript to note that those who did not visit a healthcare provider may have been healthy and did not need to see one, and it could also be an artifact of having experienced police harassment recently as well. We feel these new additions have significantly improved our manuscript. 

Results

Can the authors provide p-values for all models and regression coefficients?

In unadjusted models, there was evidence that IBPH predicted:

1) at 6M, distrust in HCP, missing more than 50% of appointments, and visiting the ED.

2) at 12M, distrust in HCP, visiting the ED.

However, in adjusted analyses, the confidence intervals for all the predictors in the logistic regressions include 1.0 (except for one predictor), which suggests non-significance of the key relationships of interest. Could the authors report all p-values? These models suggest that the hypothesized relationships are due to various covariates. Can the authors report these full models, as well?

Author response: As suggested by the reviewer, we have updated the manuscript to include the full models in the supplementary material. As we ran separate models for each of the outcomes, it would be excessive to report all of these models in the main text, but an Appendix will be included that has all covariates listed with the coefficients, p-values, and confidence intervals.

Aside from this point, this sample reported a very high rate of previous incarceration (60.1%). I wonder if this particular factor could underpin the outcomes of interest rather than incidences of IBPH. Did the authors run models with incarceration (60.1%)as a covariate or predictor?

Author response: Thank you for this suggestion. In the revised manuscript, we updated the analysis to include incarceration as a covariate in the multivariable models, and the updated analysis to include these effects along with the effect of incarceration in the multivariable models (which will be presented in the Appendix/Supplementary materials).

---

## [Decision Letter · Decision Letter 1]

5 Jun 2023

PONE-D-22-12905R1Associations between Police Harassment and Distrust in and Reduced Access to Healthcare Among Black Sexual Minority Men: A Longitudinal Analysis of HPTN 061PLOS ONE

Dear Dr. Feelemyer,

Thank you for submitting your manuscript to PLOS ONE. After careful consideration, we feel that it has merit but does not fully meet PLOS ONE’s publication criteria as it currently stands. Therefore, we invite you to submit a revised version of the manuscript that addresses the points raised during the review process.

We look forward to receiving your revised manuscript.

Kind regards,

Steve Zimmerman, PhD

Associate Editor, PLOS ONE

**Additional Editor Comments:**

We have now secured additional reviewers for you manuscript, and their comments are available below.

Although all of the reviewers agree that your work makes an important contribution, two of the reviewers have several requests for clarification and additional details, including work on the conceptualization of the issue of discrimination and the framing of the argument presented.

Could you please revise the manuscript to carefully address the concerns raised?

Reviewers' comments:

Reviewer's Responses to Questions

**Comments to the Author**

1. If the authors have adequately addressed your comments raised in a previous round of review and you feel that this manuscript is now acceptable for publication, you may indicate that here to bypass the “Comments to the Author” section, enter your conflict of interest statement in the “Confidential to Editor” section, and submit your "Accept" recommendation.

Reviewer #1: All comments have been addressed

Reviewer #2: (No Response)

Reviewer #3: (No Response)

Reviewer #4: (No Response)

2. Is the manuscript technically sound, and do the data support the conclusions?

Reviewer #1: Yes

Reviewer #2: Yes

Reviewer #3: Yes

Reviewer #4: Yes

3. Has the statistical analysis been performed appropriately and rigorously? 

Reviewer #1: Yes

Reviewer #2: Yes

Reviewer #3: Yes

Reviewer #4: Yes

4. Have the authors made all data underlying the findings in their manuscript fully available?

Reviewer #1: No

Reviewer #2: Yes

Reviewer #3: Yes

Reviewer #4: Yes

5. Is the manuscript presented in an intelligible fashion and written in standard English?

Reviewer #1: Yes

Reviewer #2: Yes

Reviewer #3: Yes

Reviewer #4: Yes

6. Review Comments to the Author

Reviewer #1: (No Response)

Reviewer #2: The study is novel and fills an important gap in the literature on the role of structural factors in explaining differential health outcomes among racial/ethnic and sexual and gender minority populations. Overall, I found the findings compelling and the authors should be commended for their intellectual rigor and theoretical framing of this question. However, I have one larger philosophical question regarding how discrimination is being conceptualized and a couple minor issues which will be more easily addressed by the authors.

1. I would recommend the authors consider being more explicit about what it is you are measuring here. You are examining exposures of racism and discrimination among Black sexual minority populations and operationalizing this as negative police encounters. However, the way this is currently written the significance of the manuscript is muted. In your methods section you indicate: ‘At the baseline and six-month visits, participants were asked about past six-month experience with police harassment motivated by racism or homophobia.’ You are measuring both racism and discrimination. However, in the text you only refer to ‘discrimination’. Please add racism e.g., racism and discrimination in those instances where discrimination is used. APA delineates between discrimination and racism: Racism is a form of prejudice that assumes that the members of racial categories have distinctive characteristics and that these differences result in some racial groups being inferior to others. Racism generally includes negative emotional reactions to members of the group, acceptance of negative stereotypes, and racial discrimination against individuals; in some cases it leads to violence. Discrimination refers to the differential treatment of the members of different ethnic, religious, national, or other groups. Discrimination is usually the behavioral manifestation of prejudice and therefore involves negative, hostile, and injurious treatment of members of rejected groups. https://www.apa.org/topics/racism-bias-discrimination

2. Also, I draw your attention to the following excerpt from an article by Lisa Bowleg on how to write about racism. She writes: ‘The language that researchers use shapes virtually every aspect of the research process. Consequently, these are not simply pedantic concerns. If researchers conceptualize “race” rather than social processes based on race (e.g., racism, racial trauma) as the source of health inequities, then every decision (e.g., the operationalization of variables, measures, hypotheses, analyses, interpretation of results, and the potential for intervention) hinges on this choice. It also detracts attention from more fundamental and modifiable factors such as those shaped by structural racism (e.g., occupation, household composition) that often provide greater explanatory power for racialized health inequities such as COVID-19 (coronavirus disease 2019) than “race” (see, e.g., Selden & Berdahl, 2020). Avoid euphemisms and other linguistic tropes that color-blind or otherwise erase the structural roots of racialized health inequities.’ (Please see for full article: https://journals.sagepub.com/doi/full/10.1177/10901981211007402). With that in mind, I wonder if the term ‘identity-based police harassment’ erases the state-sanctioned racialized violence aspects of these encounters and neutralizes its structural roots thus detracting from the fundamental thrust of your findings. Please (re)consider utilizing another term that is more precise in naming what this variable is measuring. In the survey it was referred to as ‘police harassment motivated by racism or homophobia’. Perhaps racialized police harassment and homophobia or some other appropriate descriptor?

3. In the introduction on p. 3, the authors wrote: ‘Dual minority status—being both racial/ethnic minority as well as sexual minority—is thought to play a role in psychosocial vulnerability and adverse mental health outcomes, while network and structural factors also are implicated (7). Given the substantial health challenges experienced by Black sexual minority groups, there is a critical need to ensure adequate access to healthcare to ensure screening, and treatment and prevention to address mental health, substance use and related infectious disease risk.’ This second sentence is where you want to really hone in on what it is that you are hypothesizing. However, something seems to be missing. Suggested change: Given the enormous health disparities faced by Black sexual minority groups, there is a critical need to understand the impact of racism and discrimination on health outcomes to ensure adequate access to healthcare including screening, prevention, and treatment to address mental health, substance use and related infectious disease risk.’ But of course feel free to modify the language as you see fit.

4. In the methods section where you define Identity Based Police Harassment Exposure measure, please include an example of the question(s) from the survey.

5. In the methods section, the authors wrote: We coded their responses into a three level ordinal exposure indicating increasing frequency of police harassment due to either racism or homophobia: level 0 indicates no such harassment at either time period (i.e., baseline or the six month follow-up visit), level 1 indicates that the participant reported IBPH at the baseline visit or at the six month follow-up visit, and 8 level 2 indicates that the participant reported IBPH at both the baseline and six month follow-up visit. I’m curious about why you measured IBPH. Were respondents asked to indicate exposure by a yes/no response or where they asked to provide a frequency. Because the way it’s currently measured it may not fully capture the frequency or severity of the exposure. Let’s imagine a hypothetical case that respondents were asked to indicate how many exposures they had in the last 6 months. Respondent 1 responded 6 at baseline and 1 at 12 months, whereas Respondent 2 indicated 12 and zero. Based on the way you measured it, Respondent 1 would be placed in category 3 (7 exposures) and Respondent 2 in category 2 (12 exposures), although Respondent 2 had a greater number of IPBHs. Perhaps I’m misinterpreting this, but it seems at the very least the measure is imprecise and should be addressed in the limitations section.

6. In Table 1, you use different language to label IBPH at 6mo vs 12mo. For example, at 6mo you use ‘no police harassment’ while at 12 mo ‘no IBPH’. Please select one or the other for consistency.

7. On p. 19 you note you controlled for a number of risky behaviors. I would recommend using risk factors instead of ‘risky behaviors’ as it has the potential to further stigmatize an already vulnerable population. Also, I’m not sure if I would include ‘demographics’ as a risky behavior. Please check.

Reviewer #3: This paper addresses an important and novel research question about the relationship between police harassment and healthcare access among a multiply marginalized subpopulation of adults in the U.S. The introduction is concise yet builds a solid foundation for the study's results, the methods are sufficiently described (given the additional detail about HPTN 061 published elsewhere), the analysis and results are digestible, and the discussion couches findings within the context of existing literature with notes about analytic limitations and study implications. I do not have any suggested revisions for this manuscript.

Reviewer #4: This is an interesting paper that focuses on an understudied issue - I think there is the potential for this manuscript to fill the gap in the literature, but have several suggestions I'd like the authors to address to strengthen the paper.

OVERALL

- The rampant epidemic of stigma facing minoritized populations severely impacts all components of life. While this is especially highlighted at the beginning of the Intro, I think it would be an easy framing to really tie together both IBPH and medical mistrust/healthcare access. The two points are clearly tied, but I still think the argument presented is a bit weak.

- It's a disservice to talk about stigma and multiply minoritized individuals without talking about intersectionality.

INTRODUCTION

- Page 4: need to add the word "of" - "...a study conducted in a sample 500 Black..."

- Page 4, second paragraph: I'd be explicit about what is meant by IBPH. The reader can infer that this references any minoritized identity, but I think it's worth stating here since it's crucial to the entire paper.

- Page 5, third paragraph: What about the fact that many BSMM don't even have a primary care physician? I'd think this is a bigger reason to go to the ED than intentionally avoiding their PCP.

- Page 6, first paragraph: What about the role of health insurance? A lot of these factors may be directly due to lack of healthcare insurance, especially since this study took place prior to ACA.

METHODS

- The exact wording for the IBPH question should be provided. I assume separate questions were used for race and sexual orientation? If so, why were they collapsed for this paper? Did the question specifically say "homophobia?" If so, it might not resonate with participants who did not identify as gay.

- Why wasn't sexual orientation included as a potential covariate? This seems crucial for understanding sexual identity based police harassment - likely those who identified as bisexual or straight had different experiences than those who exclusively identified as gay.

- Page 8, Covariates: Why were some of these items not actually included in the analysis? A substantial chunk of HPTN 061 participants identified as something other than cisgender men, and trans individuals are known to be harassed by police and to avoid medical care due to transphobia. Neglecting to talk specifically about this subset is a major problem.

- What was actually included in the multivariable models? Was it the list of "covariates" or was it the variables in Table 1?

RESULTS

- Why is depression mentioned here, but not in the Methods section? At the very least, the Methods should say that CES-D 10 was used.

- Definitions need to be provided for some of the less clear variables (either here or the Methods section) - specifically, health coverage, insufficient income, and unstable housing. Additionally, "STI (Any)" should indicate if it's self-report or tested, since HPTN 061 did STI testing.

- Why was one of the IBPH levels "baseline or six months?" Only experiencing IBPH at baseline or at 6 months seems worthy of distinction, especially for longitudinal analyses. I recommend separating this into two categories unless the authors have good justification.

- Table 2: what do the asterisks mean?

7. PLOS authors have the option to publish the peer review history of their article (what does this mean?). If published, this will include your full peer review and any attached files.

Reviewer #1: No

Reviewer #2: No

Reviewer #3: No

Reviewer #4: No

---

## [Author Response · Author response to Decision Letter 1]

28 Jun 2023

PLOS One Police Harassment and Healthcare Outcomes Paper

Reviewer #2: The study is novel and fills an important gap in the literature on the role of structural factors in explaining differential health outcomes among racial/ethnic and sexual and gender minority populations. Overall, I found the findings compelling and the authors should be commended for their intellectual rigor and theoretical framing of this question. However, I have one larger philosophical question regarding how discrimination is being conceptualized and a couple minor issues which will be more easily addressed by the authors.

1. I would recommend the authors consider being more explicit about what it is you are measuring here. You are examining exposures of racism and discrimination among Black sexual minority populations and operationalizing this as negative police encounters. However, the way this is currently written the significance of the manuscript is muted. In your methods section you indicate: ‘At the baseline and six-month visits, participants were asked about past six-month experience with police harassment motivated by racism or homophobia.’ You are measuring both racism and discrimination. However, in the text you only refer to ‘discrimination’. Please add racism e.g., racism and discrimination in those instances where discrimination is used. APA delineates between discrimination and racism: Racism is a form of prejudice that assumes that the members of racial categories have distinctive characteristics and that these differences result in some racial groups being inferior to others. Racism generally includes negative emotional reactions to members of the group, acceptance of negative stereotypes, and racial discrimination against individuals; in some cases it leads to violence. Discrimination refers to the differential treatment of the members of different ethnic, religious, national, or other groups. Discrimination is usually the behavioral manifestation of prejudice and therefore involves negative, hostile, and injurious treatment of members of rejected groups. https://www.apa.org/topics/racism-bias-discrimination

Author response: We thank the reviewer for this important point and reference; discrimination encompasses a broad categorization where racial discrimination would be an aspect in addition to discrimination due to sexual orientation/sexual identity. We have updated and clarified the language in the manuscript to note that discrimination is referencing both racial and sexual identity discrimination. These updates appear in the introduction and discussion sections where we refer to these concepts.

2. Also, I draw your attention to the following excerpt from an article by Lisa Bowleg on how to write about racism. She writes: ‘The language that researchers use shapes virtually every aspect of the research process. Consequently, these are not simply pedantic concerns. If researchers conceptualize “race” rather than social processes based on race (e.g., racism, racial trauma) as the source of health inequities, then every decision (e.g., the operationalization of variables, measures, hypotheses, analyses, interpretation of results, and the potential for intervention) hinges on this choice. It also detracts attention from more fundamental and modifiable factors such as those shaped by structural racism (e.g., occupation, household composition) that often provide greater explanatory power for racialized health inequities such as COVID-19 (coronavirus disease 2019) than “race” (see, e.g., Selden & Berdahl, 2020). Avoid euphemisms and other linguistic tropes that color-blind or otherwise erase the structural roots of racialized health inequities.’ (Please see for full article: https://journals.sagepub.com/doi/full/10.1177/10901981211007402). With that in mind, I wonder if the term ‘identity-based police harassment’ erases the state-sanctioned racialized violence aspects of these encounters and neutralizes its structural roots thus detracting from the fundamental thrust of your findings. Please (re)consider utilizing another term that is more precise in naming what this variable is measuring. In the survey it was referred to as ‘police harassment motivated by racism or homophobia’. Perhaps racialized police harassment and homophobia or some other appropriate descriptor?

Author response: We thank the reviewer for this important distinction in language and have modified the wording in the manuscript to “racialized and homophobia-based police harassment (RHBPH)” instead of identify based police harassment (IBPH) which is much clearer in describing the police harassment identity factors that are part of the exposure. 

3. In the introduction on p. 3, the authors wrote: ‘Dual minority status—being both racial/ethnic minority as well as sexual minority—is thought to play a role in psychosocial vulnerability and adverse mental health outcomes, while network and structural factors also are implicated (7). Given the substantial health challenges experienced by Black sexual minority groups, there is a critical need to ensure adequate access to healthcare to ensure screening, and treatment and prevention to address mental health, substance use and related infectious disease risk.’ This second sentence is where you want to really hone in on what it is that you are hypothesizing. However, something seems to be missing. Suggested change: Given the enormous health disparities faced by Black sexual minority groups, there is a critical need to understand the impact of racism and discrimination on health outcomes to ensure adequate access to healthcare including screening, prevention, and treatment to address mental health, substance use and related infectious disease risk.’ But of course feel free to modify the language as you see fit.

Author Response: We have modified this statement in the updated draft per the reviewer’s suggestion. 

4. In the methods section where you define Identity Based Police Harassment Exposure measure, please include an example of the question(s) from the survey.

Author Response: In the survey, participants were asked if they were harassed by police; if they responded yes to this question, they were asked two follow up questions including if the harassment was due to their race, or due to their sexuality. The participants were allowed to answer yes to one or both of these questions. We have added this language to the methods section where we highlight the exposure variable. 

5. In the methods section, the authors wrote: We coded their responses into a three level ordinal exposure indicating increasing frequency of police harassment due to either racism or homophobia: level 0 indicates no such harassment at either time period (i.e., baseline or the six month follow-up visit), level 1 indicates that the participant reported IBPH at the baseline visit or at the six month follow-up visit, and level 2 indicates that the participant reported IBPH at both the baseline and six month follow-up visit. I’m curious about why you measured IBPH. Were respondents asked to indicate exposure by a yes/no response or where they asked to provide a frequency. Because the way it’s currently measured it may not fully capture the frequency or severity of the exposure. Let’s imagine a hypothetical case that respondents were asked to indicate how many exposures they had in the last 6 months. Respondent 1 responded 6 at baseline and 1 at 12 months, whereas Respondent 2 indicated 12 and zero. Based on the way you measured it, Respondent 1 would be placed in category 3 (7 exposures) and Respondent 2 in category 2 (12 exposures), although Respondent 2 had a greater number of IPBHs. Perhaps I’m misinterpreting this, but it seems at the very least the measure is imprecise and should be addressed in the limitations section.

Author Response: The participants were not asked a frequency measure of the number of times the harassment occurred during the periods we measured; they were only asked if they had experienced any police harassment in the six months preceding the interview. The reviewer is correct that it is certainly possible that some participants experienced harassment more times than others; unfortunately, the questionnaire was not designed to capture this level of detail with respect to harassment due to race or sexual identity. In the limitations section we have added a note that our exposure does not capture the frequency of police harassment events. 

6. In Table 1, you use different language to label IBPH at 6mo vs 12mo. For example, at 6mo you use ‘no police harassment’ while at 12 mo ‘no IBPH’. Please select one or the other for consistency.

Author Response: We thank the reviewer for catching this inconsistency and have modified the table to use consistent language using the updated exposure title of “racialized and homophobia-based police harassment”

7. On p. 19 you note you controlled for a number of risky behaviors. I would recommend using risk factors instead of ‘risky behaviors’ as it has the potential to further stigmatize an already vulnerable population. Also, I’m not sure if I would include ‘demographics’ as a risky behavior. Please check. 

Author response: We have modified this language as well as removing the reference to any “risky behaviors” in the updated draft and refer to “factor” and “variable” instead where we present information on the statistical analysis. 

 

Reviewer #3: This paper addresses an important and novel research question about the relationship between police harassment and healthcare access among a multiply marginalized subpopulation of adults in the U.S. The introduction is concise yet builds a solid foundation for the study's results, the methods are sufficiently described (given the additional detail about HPTN 061 published elsewhere), the analysis and results are digestible, and the discussion couches findings within the context of existing literature with notes about analytic limitations and study implications. I do not have any suggested revisions for this manuscript.

Author Response: Thank you for these comments We appreciate these positive comments on our manuscript. 

Reviewer #4: This is an interesting paper that focuses on an understudied issue - I think there is the potential for this manuscript to fill the gap in the literature but have several suggestions I'd like the authors to address to strengthen the paper.

OVERALL

- The rampant epidemic of stigma facing minoritized populations severely impacts all components of life. While this is especially highlighted at the beginning of the Intro, I think it would be an easy framing to really tie together both IBPH and medical mistrust/healthcare access. The two points are clearly tied, but I still think the argument presented is a bit weak.

- It's a disservice to talk about stigma and multiply minoritized individuals without talking about intersectionality.

Author Response: Thank you for this suggestion. We have added language in the introduction to better frame this connection including “Additionally, it has been shown that for those that experience broader experiences of racism, sexual identity discrimination and harassment outside of the healthcare setting, these experiences can directly lead to mistrust in the healthcare settings (33). These negative social experiences which include RHBPH can not only shape health status, but also may influence healthcare encounters, including mistrust among doctors and clinicians in healthcare settings (34).” And “It should be noted, for instance, that excessive police violence may lead to individuals visiting the emergency department, where information would be collected on the circumstances of the injuries and may lead to further stigmatization of the individuals who experience these police related injuries (38, 39).”

INTRODUCTION

- Page 4: need to add the word "of" - "...a study conducted in a sample 500 Black..."

Author response: We thank the reviewer for the careful reading of the manuscript; we have made this correction in the updated draft. 

- Page 4, second paragraph: I'd be explicit about what is meant by IBPH. The reader can infer that this references any minoritized identity, but I think it's worth stating here since it's crucial to the entire paper.

Author response: We have made clear what IBPH refers to by updating the phrasing to now read as “racialized and homophobia-based police harassment (RHBPH)” which we think it much more clear for the reader, and what has been suggested by another reviewer as a better way to identify the exposure for the paper. 

- Page 5, third paragraph: What about the fact that many BSMM don't even have a primary care physician? I'd think this is a bigger reason to go to the ED than intentionally avoiding their PCP.

Author response: This is an excellent point; we have added this information to the introduction where we have expanded the language around the use of the ED. 

- Page 6, first paragraph: What about the role of health insurance? A lot of these factors may be directly due to lack of healthcare insurance, especially since this study took place prior to ACA.

Author response: Our sample had approximately 60% of participants reporting health insurance coverage at baseline. It is possible that the due to 40% lacking health coverage, they may rely on certain medical services. We have therefore updated the introduction to include this point. 

METHODS

- The exact wording for the IBPH question should be provided. I assume separate questions were used for race and sexual orientation? If so, why were they collapsed for this paper? Did the question specifically say "homophobia?" If so, it might not resonate with participants who did not identify as gay. 

Author response: The exposure variable was defined in the paper to replicate the exposure that was used in other papers that were part of the HPTN061 body of literature. See “Feelemyer J, Duncan DT, Dyer TV, Geller A, Scheidell JD, Young KE, Cleland CM, Turpin RE, Brewer RA, Hucks-Ortiz C, Mazumdar M, Mayer KH, Khan MR. Longitudinal Associations between Police Harassment and Experiences of Violence among Black Men Who Have Sex with Men in Six US Cities: the HPTN 061 Study. J Urban Health. 2021 Apr;98(2):172-182. doi: 10.1007/s11524-021-00526-1. Epub 2021 Apr 5. PMID: 33821426; PMCID: PMC8079523” for one example where the exposure was similarly defined. 

- Why wasn't sexual orientation included as a potential covariate? This seems crucial for understanding sexual identity based police harassment - likely those who identified as bisexual or straight had different experiences than those who exclusively identified as gay.

Author Response: We thank the reviewer for bringing up this important point. We did have one question that asked if participants had sex with men only, or if they had sex with men and women (the question asked was “do you have sex with men exclusively or do you have sex with men and women?). We included this as a covariate in the models. We have also added this variable to Table 1 where we list the demographic and other factors that were built into the multivariable models. 

- Page 8, Covariates: Why were some of these items not actually included in the analysis? A substantial chunk of HPTN 061 participants identified as something other than cisgender men, and trans individuals are known to be harassed by police and to avoid medical care due to transphobia. Neglecting to talk specifically about this subset is a major problem. 

Author Response: We controlled for transgender status in the multivariable models; we would note that transgender individuals were a small (less than 5%) part of the HPTN061 sample. We have added this information/data to Table 1. We would have liked to have done a separate analysis among transgender persons in the sample, but unfortunately did not have sufficient numbers to do this. 

- What was actually included in the multivariable models? Was it the list of "covariates" or was it the variables in Table 1? 

Author Response: All variables in Table 1 are included in the models; we have added this important point into the statistical analysis section. We have also updated Table 1 to include the breakdown of each variable/covariate with descriptive statistics for the entire sample at baseline and at the 12-month follow-up visit that was included in the multivariable model (including the variables for men who have sex with me vs men who have sex with men and women; and transgender individuals. 

RESULTS

- Why is depression mentioned here, but not in the Methods section? At the very least, the Methods should say that CES-D 10 was used.

Author Response: We thank the reviewer for noting this omission in the methods. The CES-D scale was used for the depression variable, and is now listed in the methods section with a reference now included for the scale. 

- Definitions need to be provided for some of the less clear variables (either here or the Methods section) - specifically, health coverage, insufficient income, and unstable housing. Additionally, "STI (Any)" should indicate if it's self-report or tested, since HPTN 061 did STI testing.

Author Response: We have updated the definitions for these specific variables in the methods section including lab testing for STI infection and other factors using the exact wording from the HPTN survey questionnaire. This update now appears in the methods in the section of “covariates” and is also spelled out in the Table 1 information.

- Why was one of the IBPH levels "baseline or six months?" Only experiencing IBPH at baseline or at 6 months seems worthy of distinction, especially for longitudinal analyses. I recommend separating this into two categories unless the authors have good justification.

Author response: As noted above, the exposure was measured this way to be consistent with other papers that have been published on the HPTN cohort that used this measure at these two time points in evaluating outcomes at the final (12 month) visit. 

- Table 2: what do the asterisks mean?

Author Response: The asterisks have been removed from Table 2.

---

## [Decision Letter · Decision Letter 2]

8 Aug 2023

Associations between Police Harassment and Distrust in and Reduced Access to Healthcare Among Black Sexual Minority Men: A Longitudinal Analysis of HPTN 061

PONE-D-22-12905R2

Dear Dr. Feelemyer P Jonathan,

We’re pleased to inform you that your manuscript has been judged scientifically suitable for publication and will be formally accepted for publication once it meets all outstanding technical requirements.

Kind regards,

Ubaldo Mushabe Bahemuka, Msc

Academic Editor

PLOS ONE

Additional Editor Comments (optional):

Reviewers' comments:

Reviewer's Responses to Questions

**Comments to the Author**

1. If the authors have adequately addressed your comments raised in a previous round of review and you feel that this manuscript is now acceptable for publication, you may indicate that here to bypass the “Comments to the Author” section, enter your conflict of interest statement in the “Confidential to Editor” section, and submit your "Accept" recommendation.

Reviewer #2: All comments have been addressed

Reviewer #3: All comments have been addressed

2. Is the manuscript technically sound, and do the data support the conclusions?

Reviewer #2: Yes

Reviewer #3: Yes

3. Has the statistical analysis been performed appropriately and rigorously? 

Reviewer #2: Yes

Reviewer #3: Yes

4. Have the authors made all data underlying the findings in their manuscript fully available?

Reviewer #2: Yes

Reviewer #3: No

5. Is the manuscript presented in an intelligible fashion and written in standard English?

Reviewer #2: Yes

Reviewer #3: Yes

6. Review Comments to the Author

Reviewer #2: I so nor Have any additional comments. I am accepting the manuscript without further comment. The manuscript makes an important contribution to the literature on HIV and structural discrimination.

Reviewer #3: The authors have submitted a revised manuscript that sufficiently addresses all reviewer feedback.

7. PLOS authors have the option to publish the peer review history of their article (what does this mean?). If published, this will include your full peer review and any attached files.

Reviewer #2: No

Reviewer #3: No

---

## [Editor Report · Acceptance letter]

11 Aug 2023

PONE-D-22-12905R2 

Associations between Police Harassment and Distrust in and Reduced Access to Healthcare Among Black Sexual Minority Men: A Longitudinal Analysis of HPTN 061 

Dear Dr. Feelemyer:

I'm pleased to inform you that your manuscript has been deemed suitable for publication in PLOS ONE. Congratulations! Your manuscript is now with our production department. 

Kind regards, 

on behalf of

Dr. Ubaldo Mushabe Bahemuka 

Academic Editor

PLOS ONE